

# SAR image observations of the A-68 iceberg drift

Ludwin Lopez-Lopez[1], Flavio Parmiggiani[2], Miguel Moctezuma-Flores [1], and Lorenzo Guerrieri[3]

[1]UNAM, Fac. Ingenieria, Cd. Universitaria, CDMX, 01430, Mexico
[2]CNR Institute of Polar Sciences, via Gobetti 101, Bologna, 40129, Italy
[3]INGV, via di Vigna Murata 605, Rome, 00143, Italy

**Correspondence:** M. Moctezuma-Flores (mmoctezuma@fi-b.unam.mx)

**Abstract.** A methodology for examining a temporal sequence of Synthetic Aperture Radar (SAR) images as applied to the detection of the A-68 iceberg and its drifting trajectory, is presented. Using an improved image processing scheme, the analysis covers a period of eighteen months and makes use of a set of Sentinel-1 images. A-68 iceberg calved from the Larsen C ice shelf in July 2017 and is one of the largest icebergs observed by remote sensing on record. After the calving, there was only a modest

decrease in the area (about 1%) in the first six months. It has been drifting along the east coast of the Antarctic Peninsula and it is expected to continue its path for more than a decade. It is important to track the huge A-68 iceberg to retrieve information on the physics of iceberg dynamics and for maritime security reasons. Two relevant problems are addressed by the image processing scheme presented here: (a) How to achieve quasi-automatic analysis using a fuzzy logic approach to image contrast enhancement, and (b) Adoption of ferromagnetic concepts to define a stochastic segmentation. The Ising equation is used to

model the energy function of the process, and the segmentation is the result of a stochastic minimization.

## 1 Introduction

Weather conditions and seasonal variations impose restrictions on the monitoring of Antarctica by satellite remote sensing. Continuous sunlight from December to February makes it a good period for optical image remote sensing. However, clouds,

snow and ice elements all display a similar spectral signature in both optical and thermal wavelengths. Antarctica has seven months of winter darkness, from March to September. During the Antarctic night, both SAR and infra-red images can monitor ice coverage, however, cloudy weather makes infra-red observation impossible. The scatterometer is an alternative instrument, but because of its low spatial resolution, it can only give rough estimations of large icebergs. Consequently, continuous monitoring of Antarctica can only be carried out by SAR imaging systems. This paper gives an example of Antarctic monitoring by

analysing some elements of the drifting trajectory of the A-68A iceberg using Sentinel-1 SAR data.

The fracture of the Antarctic Larsen C ice shelf occurred in 2017 between July 10th and 12th, with a loss of some 5,800 km$^2$ corresponding to about 12% of the entire shelf area. The giant calved iceberg was named "A-68" by the US National Ice Center (USNIC). Later it broke apart and the largest chunk was named A-68A. It is the sixth-largest recorded iceberg, and at





present, it is the largest iceberg in the world. Because of its size, an iceberg like A-68A can have a life of several years. Iceberg

drifting patterns constitute a risk for navigation and shipping routes. Satellite remote sensing imagery can provide the tool for mapping iceberg trajectory progression.

In iceberg monitoring by remote sensing there are two basic objectives: iceberg detection and iceberg drifting forecast. For iceberg detection, a hierarchical object-based segmentation is applied to a set of geometrical parameters of ENVISAT/ASAR images (Mazur et al., 2017). The radar altimeter is an alternative instrument and in (Tournadre et al., 2016) the signatures of

icebergs in waveform space are analysed by threshold criteria to parametrize iceberg distribution. In (Scheick et al., 2019), a machine learning technique is applied to mask clouds in multispectral Landsat images. Then, iceberg detection is performed by threshold criteria being careful to notice the radiometric contrast between icebergs and the surrounding open sea. Using Sentinel-1 SAR and CryoSat-2 SIRAL data, Han et al. (Han et al., 2019) describe the topological evolution of iceberg A68 and investigate the effects of environmental forces over a period of 18 months. A review of the remote sensing of the cryosphere

and processing techniques for sea ice can be found in (Meier and Markus, 2014; Zakhvatkina et al., 2019). For more than 45 years passive microwave images have been used for monitoring polar regions. The natural molecular interaction of the scene elements produces an electromagnetic radiation which can be used by passive microwave (PMW) sensors to discriminate the electromagnetic signatures of water, snow cover and ice extent (Thomas, 1986). Main applications of PMW images are sea-ice concentration analysis (Ivanova et al., 2014; Kern et al., 2019), thin ice studies (Mäkynen and Similä, 2015), sea-ice extent and

ice-edge location (Meier and Stroeve, 2008), and sea ice production (Preußer et al., 2019). However, the emitted radiation is very low, and consequently the passive energy must be compiled over large regions. With a swath width of 1450 km, the spatial resolution of the Advanced Microwave Scanning Radiometer 2 (AMSR2) ranges from $5 \times 3$ km to $62 \times 35$ km. Thus, PMW instruments are appropriate to observe large regions and can give only a rough estimation when applied to iceberg monitoring.

For iceberg and ice tracking forecasting, an unmanned aerial vehicle platform was used to analyse thermal video (Leira et al.,

2017). A set of dynamic forecasts was obtained using GPS trackers positioned on icebergs in (Yulmetov et al., 2016). Based on Sentinel-1 images in (Demchev et al., 2017) non-linear diffusion filtering reduces the speckle noise, and features are detected in a non-linear multiscale space representation; nearest-neighbour matching reveals the connections between the extracted features, these being the basis for sea ice drift tracking. In another study (Muckenhuber et al., 2016), the Sentinel-1 SAR image resolution is reduced by a spatial average operation to decrease speckle influence. Then, sea ice tracking is performed using a

scale-invariant feature transform algorithm. In a set of ENVISAT/ASAR images, after morphological characterization by pixel-based segmentation, tracking is performed using ocean current data (Collares et al., 2018). In (Wesche and Dierking, 2016), a drift model makes use of wind predictions for estimating positions and trajectories of icebergs observed in ENVISAT/ASAR images. More complete delineations, such as the statistical, kinematic and dynamic models, require hydro-meteorological data and both atmosphere and ocean circulation models (Diansky et al., 2018). In general, modelling the interacting forces is a very

complex task (Andersson et al., 2016; Bigg et al., 2018).

With regard to the image processing domain, SAR reconnaissance capabilities are limited by the peculiar behaviour of radar imaging; indeed, basic problems, such as the irregular image contrast and the multiplicative degradation by speckle noise are still a challenge. Pixel-based techniques, such as *K*-means, Fuzzy *C*-means, minimum distance criteria and normalized multi-





band indexes are well suited for optical and multi-band images, but their algorithmic performance is limited by the random
nature of the SAR data (Maître, 2010).

For this reason, in this paper, the stochastic process theory is taken into account. For modelling the spatial interaction of
pixel data, a model based on concepts of statistical ferromagnetism appears promising. Two relevant problems are addressed
by our image processing technique: (a) Low-level fuzzy logic image contrast enhancement, which was derived from medical
image analysis, and (b) A segmentation algorithm which considers the random behaviour of the SAR imagery for merging
contextual data. A processing scheme was then implemented which consists of the following steps: (1) Contrast enhancement;
(2) Stochastic segmentation, and (3) Measurement of the drift trajectory.

## 2 Material

This study is based on a set of twelve Sentinel-1 Extra Wide Swath Ground Range Detected (S1 EW GRD, 400 km swath, 20
x 40 m spatial resolution) images at Level-1 in HH polarization; the images were acquired from 22 July 2017 to 26 January
2019 and their geographical coordinates range from latitude 66° S to 69° S and from longitude 57° W to 63° W. After retrieval
from the ESA Scientific Data Hub, the images were remapped onto a regular grid in stereo-polar projection with a pixel size
of $200{\times}200$ m. The scene size is $400{\times}400$ km. Figure 1 shows the image corresponding to 22 July 2017, just a few days after
the calving event. In Antarctica, most icebergs are created by the calving of ice shelves and glacier tongues. The flat plateau
top appearance is a characteristic feature of the tabular icebergs produced in this region.

## 3 Methods

Electromagnetic variables of radar may introduce undesirable effects in the radiometric quantization so that the grey-level
distribution displays a histogram with saturation in local ranges. The subsequent effect is poor image contrast which reduces
perception capabilities. Some images in the analysed data set display this characteristic, and, for this reason, intensity transfor-
mation was included in the analysis.

### 3.1 SAR scattering

Remote sensing by SAR systems is the result of a complex electromagnetic phenomenon and the radargrammetry technique
must consider adverse variables, which may affect the function of the imaging system (Leberl, 1990). The physical manifesta-
tion of radar reflectivity is the scattering phenomenon. Diffuse and specular reflections are due to the geometric irregularities
of the surface. Other electromagnetic properties, such as the dielectric constant, permeability and conductivity complement the
scattering models. These properties modify the rate of the incident and reflected energy. Therefore, the backscattered signal
determines the radiometric signature of the scene elements.

At high latitudes, the properties of the scene elements change with time. Geophysical and climatological variables, such
as the temperature of the medium, wind speed, rain, salinity and humidity introduce dynamic fluctuations in the scattering





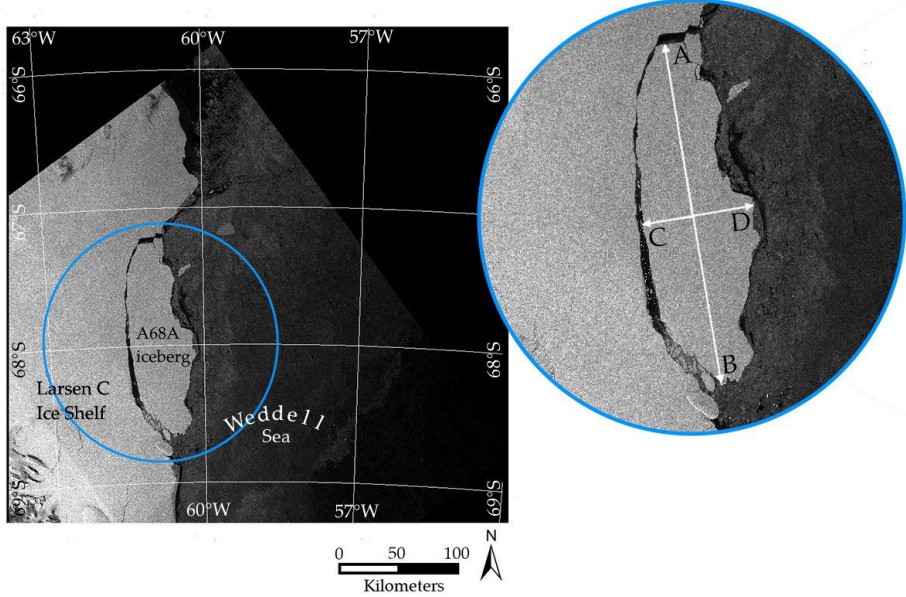

**Figure 1.** Sentinel-1 scene of A-68 tabular iceberg acquired on 22 July 2017, only a few days after the calving event. The first derived parameters of this huge tabular iceberg were: the length of its major (segment AB) and minor (segment CD) axes, which were 157.5 km and 47.3 km, respectively.

phenomenon (Xu et al., 2018). For example, new ice produces specular reflections like a thin film with a smooth appearance.
Dry snow produces a very weak reflection. Ice sheets and dry soil have a similar dielectric property (Richards, 2009). However, moisture in snow, salt and air in the ice layers all increase diffuse scattering. The variations of the dielectric property can cause two main problems in SAR image analysis: an irregular contrast of the scene elements and an overlap between the modes of the intensity histogram. These problems must be solved in the feature extraction/segmentation process and a solution to both is proposed in this section.

**3.2 Fuzzy Contrast Enhancement in the spatial domain**

Several contrast enhancement methods are currently applied in image remote sensing analysis. The main options are based on linear and non-linear formulations such as equalization, normalization, matching, logarithmic and exponential transformation functions (Solomon and Breckon, 2011). In the spatial domain, one important parameter is the dynamic range, which is defined by the smallest and the largest grey-level value of the image under analysis. To obtain an improved mapping of the grey-scale
distribution, the basic approach is to transform the dynamic range, a task which can be accomplished by fuzzy set theory (Nachtegael et al., 2003). In the case of the fuzzy histogram equalization (Kamel and Campilho, 2005), a membership function $\mu(g)$ is defined for each pixel grey-level value $g_{nm}$ at the spatial coordinates (n,m) and this is expressed by:



$$\mu(g) = \frac{g_{nm} - g_{\min}}{g_{\max} - g_{\min}}. \tag{1}$$

The terms $g_{\max}$ and $g_{\min}$ are the maximum and the minimum values of the grey-level domain. The parameter used is the

dynamic range, which is the normalization term of Eq. (1). The function $\mu(g)$ is interpreted as a homogeneity operator of the

input image luminance, or as an adapted measure of the biological perception of contrast (Pratt, 2001). Both the input image

$X$ and $\mu(g)$ have the same matrix rank.

In the original formulation of the fuzzy sets (known as type-1 fuzzy sets or T1 FSs), the inferred information is structured by

membership functions, which are a representation of the probability density function. A limitation of the fuzzification process

is that the membership functions are deterministic for a given random variable but usually the histogram of an image exhibits

mixed random variables, and this uncertainty requires additional abstraction. The next step of our scheme makes use of type-2

fuzzy sets (T2 FSs): as their membership functions become fuzzy (Mendel, 2007), we obtain a better representation of the

uncertainty and the information ambiguity of the inferred probability density function.

In this paper, the T2 FSs were the choice for implementing a contrast enhancement algorithm. Equation (1) is a membership

function T1 FSs associated with a contrast enhancement procedure. Thus, a suitable T2 FSs membership function is obtained

by making Eq. (1) fuzzy. The new function is structured by assigning an interval-based set to Eq. (1) and this is accomplished

by:

$$\begin{aligned}
\mu_{\mathrm{up}} &= [\mu(g)]^{\alpha} \\
\mu_{\mathrm{low}} &= [\mu(g)]^{1/\alpha},
\end{aligned} \tag{2}$$

where $\alpha$ is a fuzzifier parameter with $0 < \alpha < 1$, and $\mu_{\mathrm{up}}$ and $\mu_{\mathrm{low}}$ are the upper and lower bounds of the T2 FSs membership

function (Tizhoosh, 2005). The $\mu_{\mathrm{up}}$ and $\mu_{\mathrm{low}}$ functions and the input image $X$ have the same matrix size. The fuzzy function

maps the input image into a grey-level transformation, and this implies multi-criteria decision making. One suitable option for

global decision making is the $t$-conorm operator, and in this paper, the adopted algebraic operator was derived from medical

image processing literature (Chaira, 2019):

$$\tilde{\mu}(g) = \frac{\mu_{\mathrm{up}} + \mu_{\mathrm{low}} + \mu_{\mathrm{up}} \cdot \mu_{\mathrm{low}} \cdot \bar{\mathrm{X}}}{\mu_{\mathrm{up}} \cdot \mu_{\mathrm{low}}(1 + \bar{\mathrm{X}}) + 1}, \tag{3}$$

where $\bar{\mathrm{X}}$ is the expected value of the input image $X$, and $\mu_{\mathrm{up}}$ and $\mu_{\mathrm{low}}$ are the T2 FSs membership functions obtained by

Eq. (2). The process begins with the ingestion of the input image $X$ into Eq. (1), afterwards Eqs. (2) and (3) are computed. The

membership function $\tilde{\mu}(g)$ maps the contrast enhancement operation.

### 3.3 Stochastic segmentation approach

SAR images are affected by multiplicative speckle degradation; therefore, even binary segmentation is not a simple task. In

an elementary polar environment description, the analysed scene is a binary field composed of open sea and ice sheet objects.





In the framework of Bayes' theory, the implementation proposal infers the relevant information from both pixel-based and locally connected pixels. The input image $X$ is a pixel lattice $S$ of N×M, where the pixel coordinates $(i,j)$ are structured by a neighbourhood system $\eta$. According to the Euclidean distance, the first and second-order system, $\eta_1$ and $\eta_2$, correspond to the 4-connected and to the 8-connected systems, respectively. A clique is a subset $C \subset S$ and it represents the primitive image structure of connected pixels or sites. For a system $\eta_2$, the associated set of cliques $C_1$ and $C_2$ are, respectively, the central pixel $X_{ij}$ and the set of pixel pairs. The spatial feature field defines a set of $n$ mutually exclusive labels $L = \{l_1, l_2, \cdots, l_n\}$. The output of the segmentation process is the variable $Y = \{y_{ij} \in L\}$.

With the term $P(X|Y)$, the Bayesian theory takes into account the probability distribution of the pixel grey-level, given the label field $Y$ and also the "a priori" information of the labelling process, i.e. the term $P(Y)$. A Bayesian maximum-a-posteriori (MAP) estimator is:

$$P(Y_l \,|\, X_{ij}) = \underset{Y_l}{\arg\max} \left( \frac{P(X_{ij}\,|\,Y_l)P(Y_l)}{P(X)} \right), \tag{4}$$

where $Y_l \in L$, and $P(X)$ is the probability of realization of the input random variable. Using a Markov random field (MRF) model, the probability terms of the MAP equation can be adapted to introduce contextual information. Once the random variable $X$ is assumed as a MRF realization, a Gibbs function models the region process of $Y$. Thanks to this concept, the terms of Eq. (4) are approached by the sum of energy functions $U \approx U(X|Y) + U(Y)$.

The term $U(X|Y)$ is considered a realization of the label set in the grey-level range, and, in this paper, the conditional modes are expressed by Gaussian functions $U(X|Y) = \ln(\sqrt{2\pi}\sigma_i) + (X - \bar{Y}_i)^2(2\sigma_i^2)^{-1}$, where $\bar{Y}_i$ is the mean, and $\sigma_i^2$ is the variance of the label $Y_i$. The MRF theory is based on statistical physics (SIGELLE and RONFARD, 1992), and in this paper for introducing the function $U(Y)$, the Ising model (Ibe, 2013) was implemented following a ferromagnetic interpretation of the random process. The cardinality of the sites $\sigma_i$ is specified through the local label arrangement of $Y$:

$$U(Y) = -\alpha M \sigma_i - \beta \sum_{ij} \sigma_i \sigma_j. \tag{5}$$

In a ferromagnetic reading, $\alpha$ is a characteristic of the involved element, $M$ is a supplementary magnetic field, and $\beta$ is the magnetic condition of the material. The effect of $M$ is to induce alignment of the ferromagnetic elements in the direction of the field of $M$. The $\beta$ parameter indicates the interactive magnetic forces of adjacent sites. The magnetic attractive case occurs when $\beta > 0$. The joint effect of $M$ and $\beta$ is to produce states of low energy, and in the case of a segmentation process, to generate homogeneous label configurations. Thus, the resulting $U$ function is driven by:

$$U = \sum_{c_1 \in C} U(X|Y) + \sum_{c_2 \in C} U(Y). \tag{6}$$

To find the optimal estimate of the label field $L$, a numerical minimization of $U$ is needed. As Eq. (6) is a non-convex function displaying different local minimal energy states (zero slop intervals), in order to induce progressive low energy configurations, a simulated annealing scheme was implemented. To obtain further adjustments in the local energy array, thus allowing to reach





a global minimum state, the Gibbs sampler criterion (Chatelain et al., 2011) was applied. A homogeneous grouping of the pixels is obtained at the end of the recursion.

## 4 Results

### 4.1 Fuzzy Contrast Enhancement

In order to evaluate its performance, the applied fuzzy algorithm was compared with alternative contrast solutions: (a) the contrast limited adaptive histogram equalization (CLAHE) (Zuiderveld, 1994), and (b) the exponential grey-scale transformation. The SAR image with an acquisition date of 13 December 2017 was selected as the test image as it displays a deficient contrast. Figure 2 shows: (a) The full input SAR image; (b) A selected window of $700 \times 700$ pixels of the input image; (c) Result of the CLAHE algorithm; (d) The exponential grey-scale transformation, and (e) Result of the applied fuzzy algorithm.

The fuzzifier parameter was fixed to $\alpha = 0.6$. It is observed in (c) and (d) a regular distribution of the contrast. Both open-sea and non-open-sea elements are ambiguous regions and cannot be precisely defined even by visual inspection. The results are therefore unsatisfactory for a subsequent segmentation stage. In (e) the dark regions are slightly brighter, and the bright regions are brighter as well. The fuzzy function maps the input image to a grey-level transformation, in agreement with the visual perception of contrast. Hence, the contrast of the sea-ice elements is enhanced.

### 4.2 Stochastic segmentation


The energy term $U(X|Y)$ requires the mean and variance of the Gaussian modes. The set of parameters was obtained by manually training windows over the observed ice sheet and non-ice sheet regions. In terms of the Ising model, the parameters of Eq. (5), were fixed to $\alpha = 0.3$, $M = 1$ and $\beta = 0.35$. The variables $\sigma_i$ and $\sigma_{ij}$ are the one-site clique and the two-site cliques, respectively. The parameters $\alpha$, $M$ and $\beta$ were fixed by experimental evidence. A modest contribution is expected by

the pixel-based analysis and, for this reason, the information of the pixel $\sigma_i$ was given using $\alpha < 0.5$ and a $M$ value equal to 1. The parameter $\beta$ is important because $\beta \approx 0$ produces under segmentation while $\beta \geq 1$ over segmentation. Thus, an appropriate domain is $0.3 < \beta < 0.4$. The simulated annealing process requires a numerical simulation: in consideration of its parameters, the number of iterations was fixed at 40. The derived variance of the A-68A grey-level ranges from 100 to 400, which means a transformation of the pixel region process. Consequently, an overlap is observed between the modes of the

intensity histogram, and this is a basic problem in SAR image segmentation. To tackle this problem, a contextual second-order neighbourhood model and the Ising model are necessary to MRF segmentation. Two examples of the result obtained by the proposed segmentation algorithm are shown in Fig. 3; Fig. 3(a) shows the iceberg detection of the image of 13 December 2017 while Fig. 3(b) the result corresponding to 18 January 2018. The detected iceberg shape is displayed in white and, for a better visualization, the whole SAR scenes are used.



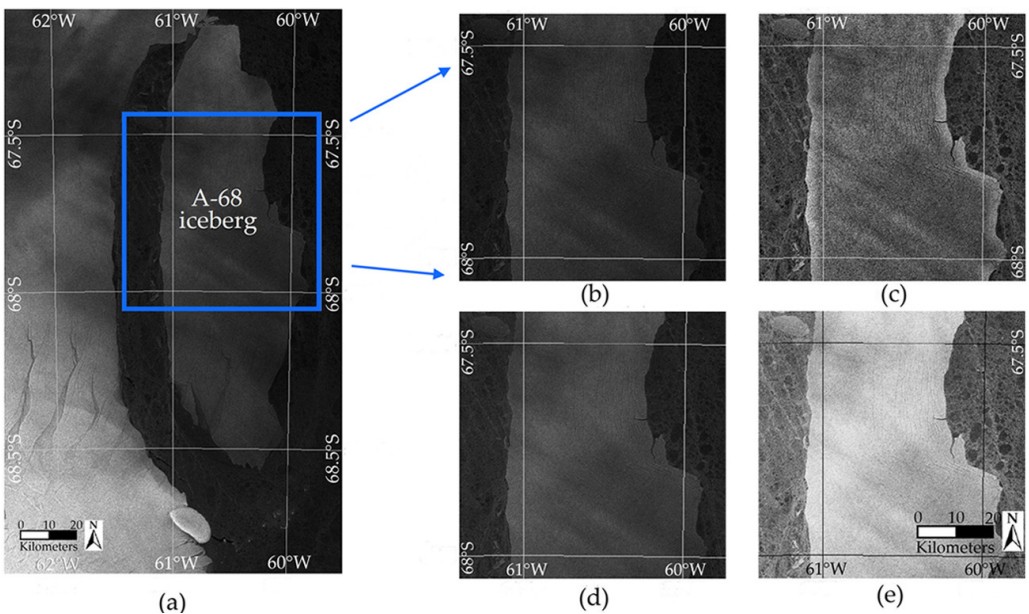

**Figure 2.** Comparison of contrast enhancement algorithms: (a) Overview of the input SAR image of 13 December 2017; (b) A window of the SAR input image; (c) Result of the CLAHE algorithm; (d) The exponential grey-scale transformation, and (e) Result of the applied fuzzy algorithm. This algorithm improves the dark and bright feature contrast. As a result, the mapping of the grey level range enhances the discrimination of the iceberg structure.

## 4.3 Measurement of the drift trajectory

Based on the segmentation result, the objects of the binary field are labelled, and the iceberg shape can be detected and extracted from the input images. Computation of its geometric parameters is now a straightforward task. For each input image, the retrieved parameters were: area, perimeter and coordinates of the centroid. For the analysed period the image acquisition dates were (1): 22 July 2017, (2): 8 August 2017, (3): 2 October 2017, (4): 13 December 2017, (5): 18 January 2018, (6): 12 June 2018, (7): 30 July 2018, (8): 11 August 2018, (9): 4 September 2018, (10): 28 September 2018, (11): 27 November 2018 and (12): 26 January 2019. Figure 4 shows the variation of the area on the left of y-axis and the perimeter on the right of the y-axis. To appreciate the long-term tendency, a curve fitting function was used: for both parameters a 5th-degree polynomial curve fits the series of data points. A decay tendency is observed in both area (blue curve) and perimeter (red curve) parameters.

Two complementary parameters are the major axis length and the rotation angle. The binary pattern of the detection is the basis for expressing the iceberg shape as a polygon with specific properties. The geometric centroid (centre of mass) of the connected iceberg pixels, i.e. concurrency point of both the major and minor axes, was derived. The two axes are marked as the segments AB and CD in Fig. 1. Taking as reference the horizontal axis and the segment defined by the centroid and point A, the rotation angle is computed in a counter-clockwise sense. Figure 5 displays the rotation angles derived from the first and



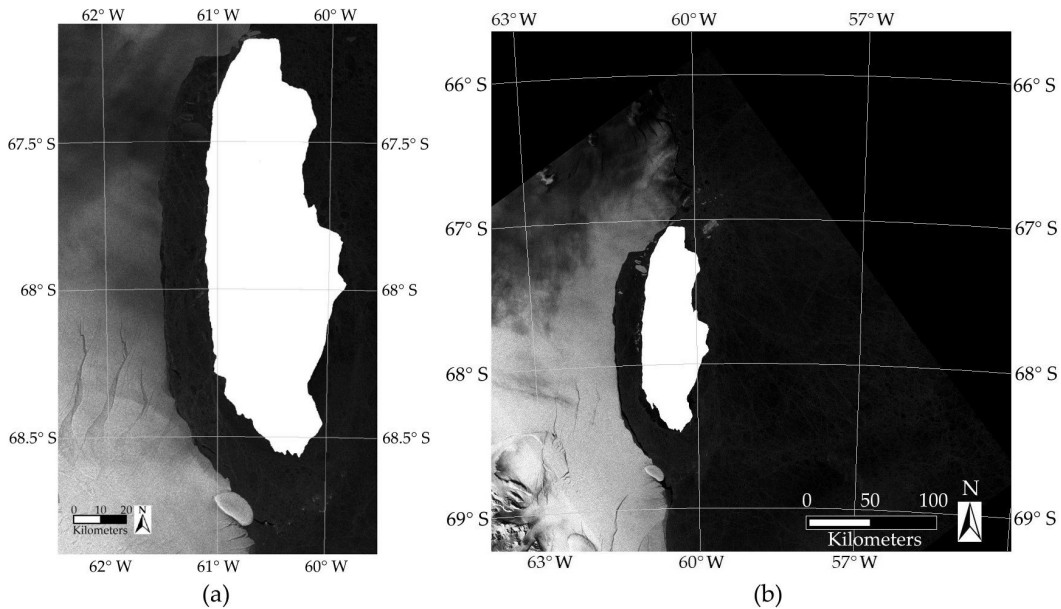

**Figure 3.** Segmentation result: (a) Image of 13 December 2017 and (b) of 18 January 2018. The detected A-68A object is displayed in the input SAR scenes. Even with the low contrast of the original scenes, the detection appears homogeneous with no spurious pixels.

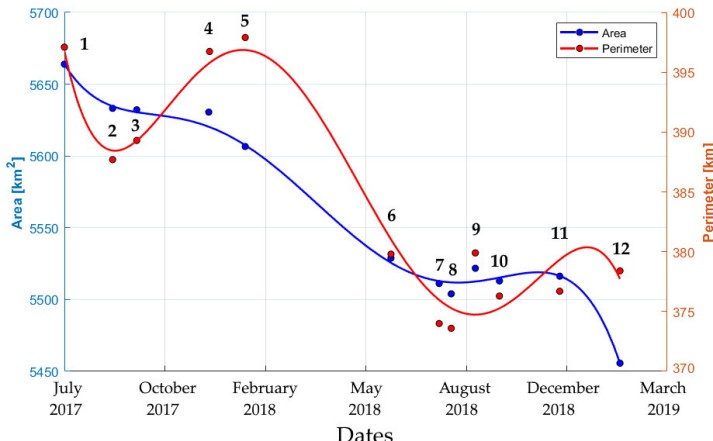

**Figure 4.** The derived time-series data of the A-68A iceberg. Based on a polynomial least squares regression, the red and blue curves display the long-term tendency of the area and perimeter parameters.

the last analysed images. Figure 6 shows the time evolution of both the rotation angle and the major axis length parameters.

For a time period of 553 days, from 22 July 2017 to 26 January 2019, Fig. 5 shows the estimated drift positions.





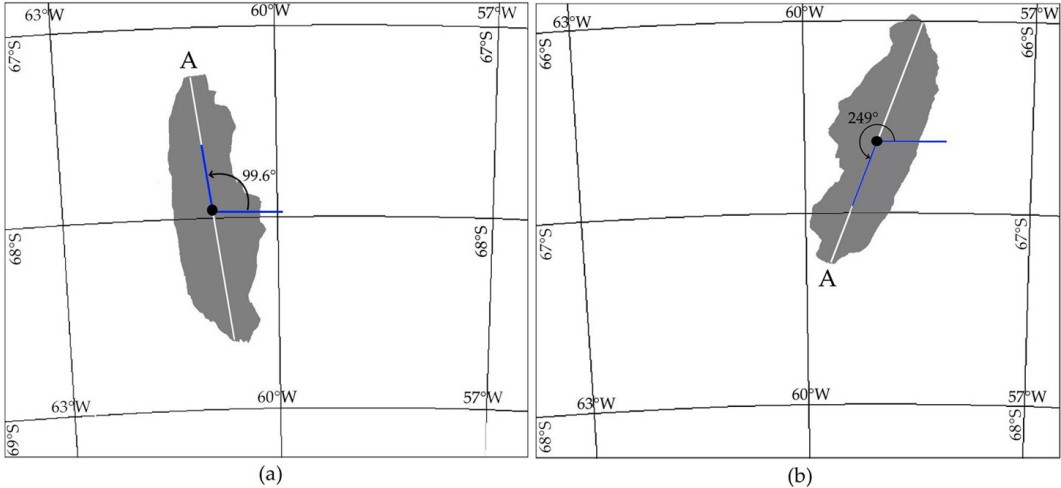

**Figure 5.** The derived rotation angle of the A-68A iceberg: (a) Image of 22 July 2017, the initial angle was 99.6°; (b) Image of 26 January 2019, the angle was 249°.

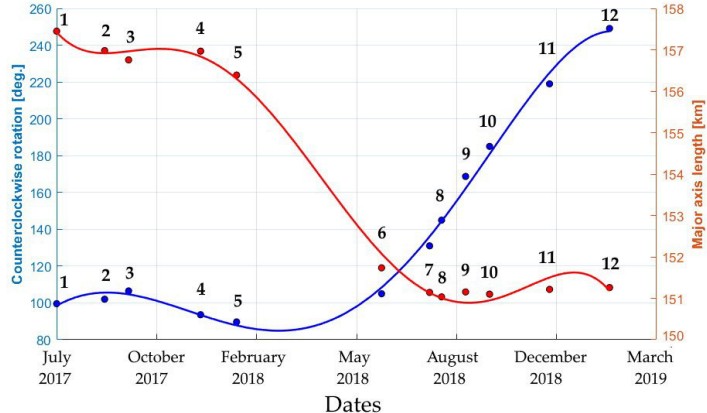

**Figure 6.** The curve of A-68A iceberg temporal displacement. Based on a polynomial least-squares regression, a 5th-degree polynomial curve fits the series of data points. The red and blue curves display the long-term tendency of rotation angle and major axis length.

## 5 Discussion

The multiplicative nature of speckle degradation introduces spurious pixel grey level values, and this statistical confusion is a basic difficulty for SAR image segmentation. To address the random nature of the SAR data, two probability abstractions provide the required information: a contextual second-order neighbourhood model and a pixel-based analysis. Therefore, the

segmented field is the result of a double segmentation model, and it was implemented using numerical optimization. Two training windows are manually fixed to derive the mean values of the ice sheet and non-ice sheet objects. This is done for each image





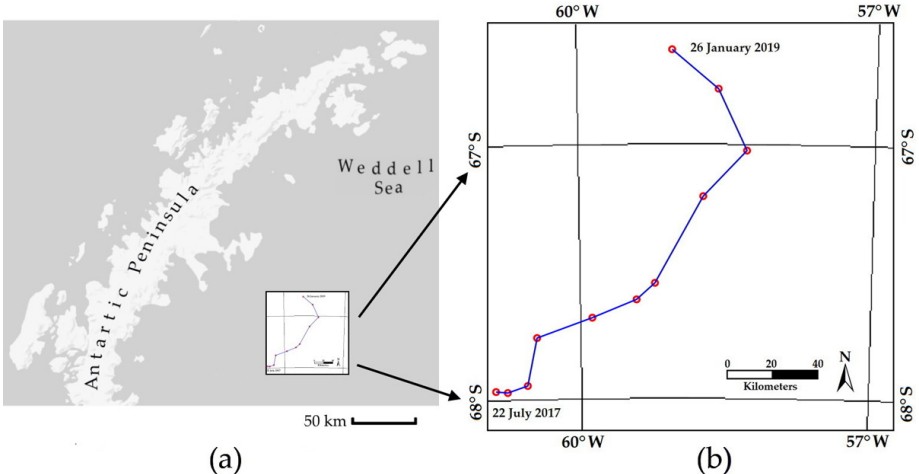

**Figure 7.** The derived sequence of the A-68A iceberg positions: (a) Overview of the derived iceberg drift, eastern Antarctic Peninsula (Google Maps ©). The blue rectangle indicates the drift area; (b) Iceberg locations. The centroid coordinates are used to display the drift trajectory.

analysed. Under-segmentation or over-segmentation is the result of an inappropriate selection of the mean values. The language used to implement the algorithms was MATLAB(R). Using a laptop with Intel(R) Core(TM) i7-7700HQ, CPU @2.8GH and 8GB Ram, the CPU time was 1.26 seconds for the fuzzy algorithm and 19.04 minutes for the stochastic segmentation. A trained
person could perform manual tracing in a reduced time, but the goal of our method is to arrive at semi-automated detection. Figure 4 shows some increases in the time series area, but this can be attributed to the attachment of sea-ice fragments. Because of the contextual analysis (to decide the class of a given pixel, its 8-neighbouring pixels are taken into account), the contours are detected with one-pixel position accuracy. Concerning the tracking application, see Fig. 7, during the year 2017 the iceberg remained near the collapse zone: the displacement was only some 30 km towards the Eastern Weddell Sea, and the area was
99% intact. The sea-floor elevation layers of the Bawden Ice Rice affected A-68A's drift movement and the iceberg did not move much during the first six months of 2018. In July 2018, it started to swing slowly in an anticlockwise direction. In the period July 2017 to August 2018, the computed mean speed was 7.2 km/month. By the end of September 2018, the rotation angle was 185° (see Fig. 6, point 10), and the speed increased to 16.8 km/month. By January 2019, the angle of the major axis was about 250°. Using the centroid data, the total displacement distance was 220.6 km. In the analysed period, a slight
reduction in the planar shape parameters was observed. The visible iceberg area reduced by about 3.7% and the major axis length by 3.9%. Melting, breakup and forced motion are consequences of the iceberg-environment interaction; main driving force arises from the surrounding ocean with some atmospheric contributions. Large icebergs last for several years and the gravitational force may introduce topological changes. The gravitational force pushes outward the iceberg mass, and, over the years, the cumulated effect produces a decrease in thickness and an increase in iceberg length (Bigg, 2015). The influence of
these elements is out of the scope of this paper. In the last analysed image, the A-68A iceberg was approaching the marginal



zone of the Antarctic Circle. At this point, the coastal current is expected to be the driving force of its displacement. Moving in the direction of the Scotia Sea, the iceberg must still travel about 400 km to reach the northernmost point of the Antarctic Peninsula.

## 6    Conclusions

A methodology is proposed for the analysis of a temporal sequence of SAR images. Two fundamental problems in the remote sensing domain are the irregular image contrast and the mixed multimodal class distribution. This paper takes into account the image uncertainty for proposing the combined use of fuzzy logic and of a ferromagnetic approach which models overlapping class intervals. A pre-processing stage implements a fuzzy contrast enhancement in the spatial domain. In the fuzzification process, a set of image features define the membership functions whose domain and range are a rough fit to the image feature

histogram. The concepts of ferromagnetic theory were chosen to define a stochastic segmentation method. In ferromagnetic theory, the effect of an external magnetic field is to induce alignment of the ferromagnetic elements; this, in the case of a segmentation process, simulates the magnetic attractive force by generating local homogeneous pixel configurations. The Ising model and the Bayes equation were the basis for implementing the spatial pixel interaction. The derived binary field is the result of a stochastic minimization process. Because of the scene size and of the recursive nature of the optimization algorithm, the

computational requirements of the MRF segmentation are computation-intensive. The final analysis result shows the movement of the A-68A iceberg over a time period. Due to its colossal size, small variations in area, perimeter and major axis length parameters were observed. Up to 26 January 2019, the detected area was 96% of its original size. The surrounding ice in the winter season, wind patterns and sea-floor elevation layers cause irregular displacements and variant iceberg velocities, but the dominant direction seems to be towards the Eastern Weddell Sea. The main contribution of this paper is in the image

processing domain with an application to the tracking of the A-68A iceberg. Ancillary information such as meteorological data, ocean currents, wind speed, temperature and geomorphology of the seabed was not available for this study, but the proposed methodology can be integrated to perform dynamic modelling.

*Data availability.*  Data are available upon request from: mmoctezuma@fi-b.unam.mx

*Author contributions.*  Conceptualization, L.L, F.P., M.M. and L.G.; Methodology, L.L., F.P., M.M. and L.G.; Validation, L.L., F.P. and L.G.;

Formal analysis, L.L., F.P., M.M. and L.G.; Data curation, L.G.; Writing–review and editing, L.L., F.P., M.M. and L.G.; Visualization, L.L., F.P., M.M. and L.G.; Supervision, F.P.;

*Competing interests.*  The authors declare that they have no conflict of interest





*Financial support.* This research was funded partially by Universidad Nacional Autónoma de México and Consejo Nacional de Ciencia y Tecnología through scholarship grant number 099598462.



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
