# Peer review of "SAR image observations of the A-68 iceberg drift"

_The Cryosphere, 2020_

## Referee Comment (RC1) · Anonymous Referee #1 · 18 Aug 2020

Review of L. Lopez-Lopez, F. Parmiggiani, M. Moctezuma-Flores, and L. Guerrieri "SAR image observations of the A-68 iceberg drift".

This paper presents an approach for automated detection and delineation of a very large iceberg from Sentinel-1 SAR images. The approach was applied to a series of 12 Sentinel-1 images (July 2017 - January 2019) acquired over the A-68 iceberg calved from the Larsen C Ice Shelf in July 2017. Several physical properties of the iceberg (such as area, perimeter, rotation, drift) as functions of time were derived from SAR. Even though some value of the proposed approach was demonstrated, I believe that the paper in its current state is not suitable for publication, and much more work needs to be done before the manuscript could be considered for publication again as outlined below.

1. It appears that a very similar approach applied to the same iceberg A-68 was already published in [r1]. Three authors in [r1] are the same authors as in the present paper. I wonder why [r1] was not cited in the present paper? Another paper [r2] also uses a similar segmentation approach (although the iceberg is different). Therefore, the novelty of the presented approach is very difficult to assess.

2. In [r3] (which was cited by the authors) a similar analysis of the same A-68 iceberg from Sentinel-1 images was conducted. But in [r3] the iceberg was manually delineated from 78 Sentinel-1 images acquired between 22 July 2017 and 29 November 2018. This is almost the same time period as the time period used in this study (July 2017 - January 2019). Therefore, the time-series results for the iceberg physical properties presented in this paper are not novel.

3. In [r3], a much larger number of Sentinel-1 SAR images (72) was used compared to the present study (12). Why all these 72 SAR images available were not utilized in this study, particularly given the fact that the proposed approach is automated? Furthermore, results in [r3] seem to be more reliable compared to the present study, as the iceberg area in [r3] was manually delineated in each SAR image. The iceberg parameters (such as its area, rotation) manually derived in [r3] could serve as a good ground-truth information for the proposed approach in this study.

4. If I visually compare Figure 4 of this study against Figure 3(a) from [r3] (showing the iceberg area versus time), I can see that the iceberg areas presented in this study are considerably lower compared to the iceberg areas reported in [r3]. How the authors can explain that difference given the fact that the results in [r3] are manually derived?

5. It is not clear how well the algorithm performs in summer time (when the iceberg might look dark and similar to the background due to melt).

6. The authors used only HH channel for iceberg detection and delineation, but HV channel is also available in Sentinel-1. I strongly suggest that HV channel should be also included in the algorithm. The authors should investigate in detail if/how the additional HV channel is capable to improve the algorithm performance.

7. The algorithm performance should be compared against other approaches such as [r4] (not cited by the authors).

8. In Introduction section, it is not clear why the authors describe passive microwave remote sensing of sea ice instead of focusing on SAR and the existing SAR image processing approaches with respect to detecting large icebergs.

9. Even though I am not a native speaker, I believe that the language of the paper should be substantially improved. There is quite a few grammatical errors, confusing sentences, and inaccuracies in the paper.

Technical corrections:

There is a lot of confusing sentences, grammatical errors, and inaccuracies in the paper. However, I believe that my major comments (stated above) should be addressed first, before I start going deeper into the technical details.

References:

[r1] Parmiggiani, F.; Moctezuma-Flores, M.; Guerrieri, L.; Battagliere, M.L. SAR analysis of the Larsen-C A-68 iceberg displacements. Int. J. Remote Sens. 2018, 39, 5850–5858.

[r2] Moctezuma-Flores, M.; Parmiggiani, F. Tracking of the iceberg created by the Nansen Ice Shelf collapse. Int. J. Remote Sens. 2017, 38, 1224–1234.

[r3] Han, H., Lee, S., Kim, J.-I., Kim, S. H., and Kim, H.-C.: Changes in a Giant Iceberg Created from the Collapse of the Larsen C Ice Shelf, Antarctic Peninsula, Derived from Sentinel-1 and CryoSat-2 Data, Remote Sensing, 11, 2019.

[r4] Silva, T.A.M.; Bigg, G.R. Computer-based identification and tracking of Antarctic icebergs in SAR images. Remote Sens. Environ. 2005, 94, 287–297

---

## Author Comment (AC1) · 7 Sep 2020

Dear Sirs, Thank you very much for your positive assessment and constructive suggestions. The replies to all the comments term by term are given as follows

1. It appears that a very similar approach applied to the same iceberg A-68 was already published in [r1]. Three authors in [r1] are the same authors as in the present paper. I wonder why [r1] was not cited in the present paper? Another paper [r2] also uses a similar segmentation approach (although the iceberg is different). Therefore, the novelty of the presented approach is very difficult to assess. Reply: Our aim was to extend on a longer period the analysis already started in [r1]. Two basic problems in the analysis of SAR images are the non-uniform grey level distribution and the speckle

[Figure]

noise random behaviour. Our proposal is based on a fuzzy-logic contrast enhancement technique and a full developed Ising model (the information of 1st and 2nd order cliques is taken into account). These algorithmic elements are not contained in [r1, r2]. The implementation of the Markov random field theory requires an analytic model, which by probability inference is formally stated by the Bayes' theorem. Thus, the Bayesian framework is a common tool with [r1, r2], but is required for the introduction of the data-driven functions. References [r1, r2] are now cited.

2. In [r3] (which was cited by the authors) a similar analysis of the same A-68 iceberg from Sentinel-1 images was conducted. But in [r3] the iceberg was manually delineated from 78 Sentinel-1 images acquired between 22 July 2017 and 29 November 2018. This is almost the same time period as the time period used in this study (July 2017 - January 2019). Therefore, the time-series results for the iceberg physical properties presented in this paper are not novel. Reply: Our aim was to propose a more automated SAR image processing procedure. We do not believe it was necessary to use 78 images to track the path of A-68 iceberg in the study period. Regarding the significance of our time-series outcomes, please, see the answer to question 4.

3. In [r3], a much larger number of Sentinel-1 SAR images (72) was used compared to the present study (12). Why all these 72 SAR images available were not utilized in this study, particularly given the fact that the proposed approach is automated? Furthermore, results in [r3] seem to be more reliable compared to the present study, as the iceberg area in [r3] was manually delineated in each SAR image. The iceberg parameters (such as its area, rotation) manually derived in [r3] could serve as a good ground-truth information for the proposed approach in this study. Reply: We do not appreciate and do not trust the manual detection of the iceberg contour, or the manual derivation of iceberg parameters (area, rotation). But concerning the manual tracing and ground-truth data remarks, please, see the answer to question 4.

4. If I visually compare Figure 4 of this study against Figure 3(a) from [r3] (showing the iceberg area versus time), I can see that the iceberg areas presented in this study are

considerably lower compared to the iceberg areas reported in [r3]. How the authors can explain that difference given the fact that the results in [r3] are manually derived? Reply: The polar stereographic projection and the pixel sampling were performed using TeraScan, https://www.seaspace.com/software-products/. We wish to stress the experience gained in the 30-year use of TeraScan software. But, are our results reliable? How to validate our results? The measure of the goodness of a segmentation method is one of the most difficult tasks in remote sensing. On this subject, we have two observations. A: Our results are based on the iceberg area estimation. Just after the calving event, for the image of 22 July 2017 we compute the area of the A68A iceberg in 5663.9 km2, but taking into account other derived fragments, the A68B iceberg and a neighbouring small iceberg, the whole estimated area is of 5760.94 km2. This corresponds to the estimation of 5800 km2 provided by the UK Midas Project in July 2017 (http://www.projectmidas.org/). Now, in section 4.3 we include a clarifying note "The applied analysis does not take into account the A68B iceberg". This observation is also included in the caption of Figure 4. B: For each analysed image, we derive a binary mask where the iceberg area is computed. To show the reliability obtained, below we include a segmentation result (see Appendix A). In the attached figures, we show a) the input SAR image of 4 September 2018 and b) the segmentation result. In the input image, it is observed a well open-sea/ice contrast, such that a manual tracing could be "easily" performed. In c) - f), four enlarged windows show details of our processing result, where the detected iceberg contour is displayed in blue. Given the suitable contrast of the input image, we consider that our estimate is equivalent to that obtained by manual delineation. Then, we are pretty sure of our outcomes.

5. It is not clear how well the algorithm performs in summer time (when the iceberg might look dark and similar to the background due to melt). Reply: We carefully avoided using summer images in order to have a better contrast between ice and sea. A multi-season analysis can be the subject of a supplementary publication.

6. The authors used only HH channel for iceberg detection and delineation, but HV

channel is also available in Sentinel-1. I strongly suggest that HV channel should be also included in the algorithm. The authors should investigate in detail if/how the additional HV channel is capable to improve the algorithm performance. Reply: Our group has a long experience in the SAR analysis of sea ice images as several publications can confirm (see Appendix B); based on our experience we decided that the HH channel is the best one for sea-ice detection. Besides, it has been demonstrated that HH is the more efficient polarization for sea ice classification (see Sentinel-1 User Handbook, SP-1322/1, 978-92-9221-418-0, Pages 62- 63).

7. The algorithm performance should be compared against other approaches such as [r4] (not cited by the authors). Reply: We have checked reference [r4]. In summary, it consists of: "The multiresolution filter of FjoCrtoft et al. (1997) is used to calculate an edge map, which is then segmented by the watershed algorithm (Beucher & Lantuejoul, 1978; Soille, 2002; Vincent & Soille, 1991). The basin dynamic method (Grimaud, 1992) was chosen to limit oversegmentation, where the threshold is chosen with the help of a contour dynamics map (Najman & Schmitt, 1996; Schmitt, 1998). As in FjoCrtoft (1999b), a merging step is applied at the end to correct oversegmentation and make the algorithm less sensitive to the choice of basin dynamics. Here we use a simple heuristic merging rule.". Then, it comprises a set of several linked algorithms. However, we note the lack of information on various criteria and parameters. We consider that it is not possible to reproduce exactly the referred algorithm. In order to follow your observation, we alternatively include a comparison with a semi-automatic segmentation algorithm: the k-means algorithm (see Fig. 3).

8. In Introduction section, it is not clear why the authors describe passive microwave remote sensing of sea ice instead of focusing on SAR and the existing SAR image processing approaches with respect to detecting large icebergs. Reply: According to your observation, the sentence referring to passive microwave remote sensing was removed.

9. Even though I am not a native speaker, I believe that the language of the paper

should be substantially improved. There is quite a few grammatical errors, confusing sentences, and inaccuracies in the paper. Reply: Before submission, the paper was revised by a mother tongue English, actually Scottish, text editing expert. We would appreciate if the reviewer could list some of the confusing sentences or inaccuracies.

References: [r1] Parmiggiani, F.; Moctezuma-Flores, M.; Guerrieri, L.; Battagliere, M.L. SAR analysis of the Larsen-C A-68 iceberg displacements. Int. J. Remote Sens. 2018, 39, 5850–5858. [r2] Moctezuma-Flores, M.; Parmiggiani, F. Tracking of the iceberg created by the Nansen Ice Shelf collapse. Int. J. Remote Sens. 2017, 38, 1224–1234. [r3] Han, H., Lee, S., Kim, J.-I., Kim, S. H., and Kim, H.-C.: Changes in a Giant Iceberg Created from the Collapse of the Larsen C Ice Shelf, Antarctic Peninsula, Derived from Sentinel-1 and CryoSat-2 Data, Remote Sensing, 11, 2019. [r4] Silva, T.A.M.; Bigg, G.R. Computer-based identification and tracking of Antarctic icebergs in SAR images. Remote Sens. Environ. 2005, 94, 287–297

**APPENDIX A**
Depicting the reliability of the segmentation process in a well-contrasted image

a)  Input image: 4 September 2018. Image size is 2000 x 2000 pixels.
     With a pixel size of 200m, the pixel area is 0.04 km$^2$.

[Figure]

b) Result of the MRF segmentation process. The contour of the detected iceberg form is
delineated in blue. Four windows are used to assess the reliability of our results.

**Fig. 1.** ResponseToReviewerAppendixA1

[Figure]

c) Window 1    d) Window 2

e) Window 3    f) Window 4

At a pixel level, the extracted windows show details of our contour estimate. The iceberg contour is depicted in blue, and it is one pixel wide. The attached sea-ice can introduce ambiguities in the detection, but this is also a problem in manual delineation. We consider that our estimate is equivalent to that obtained by manual delineation.

**Fig. 2.** ResponseToReviewerAppendixA2

**APPENDIX B**
**Some publications of our group**

1. P. Wadhams, G. Aulicino, F. Parmiggiani , O. G. Persson, and B. Holt, "Pancake ice thickness mapping in the Beaufort Sea from wave dispersion observed in SAR imagery", *J. Geophys. Res.- Oceans* (2018), doi: 10.1002/2017JC013003.

2. Flavio Parmiggiani, Miguel Moctezuma-Flores, Peter Wadhams, Giuseppe Aulicino, "Image Processing for Pancake Ice Detection and Size Distribution", *Int. J. Remote Sensing* (2018), doi: 10.1080/01431161.2018.1541367.

3. Parmiggiani, F., "Multi-year measurement of Terra Nova Bay winter polynya extents", *Europ. Phys. J. Plus*, doi:10.1140/epjp/i2011-11039-3, 2011.

4. Wadhams P., Aulicino G., Parmiggiani F., Pignagnoli L., "SAR Ice Thickness Mapping in the Beaufort Sea Using Wave Dispersion in Pancake Ice - A Case Study with Intensive Ground Truth", *Proc. 'Living Planet Symposium 2016'*, Prague (ESA SP-740, August 2016).

5. Giuseppe Aulicino , Peter Wadhams  and Flavio Parmiggiani, "SAR Pancake Ice Thickness Retrieval in the Terra Nova Bay (Antarctica) during the PIPERS Expedition in Winter 2017", *remote sensing* (2019), doi:10.3390/rs11212510.

6. Parmiggiani, F., "Fluctuations of Terra Nova Bay polynya as observed by  active (ASAR) and passive (AMSR-E) microwave radiometers", *Int. J. Remote Sensing*, vol. 27, 2459-2467,  2006.

7. Boccolari, M. & Parmiggiani, F.. (2017). Sea-ice area variability and trends in Arctic sectors of different morphology, 1996–2015. European Journal of Remote Sensing. 50. 377-383. 10.1080/22797254.2017.1331117.

8. Boccolari, M. & Parmiggiani, F.. (2017). Seasonal co-variability of surface downwelling longwave radiation for the 1982–2009 period in the Arctic. Advances in Science and Research. 14. 139-143. 10.5194/asr-14-139-2017.

9. Boccolari, M. & Guerrieri, Lorenzo & Parmiggiani, F.. (2014). Sea-ice distribution and variability in the East Greenland Sea, 2003-13. Proceedings of SPIE - The International Society for Optical Engineering. 9240. 10.1117/12.2066489.

10. F. Nunziata, A. Buono, M. Moctezuma-Flores, F. Parmiggiani and M. Migliaccio, "Observations of Terra Nova Bay polynya by Radarsat-2: Dual- and Single-Polarization Methods", 3° International Forum on Research and Technologies for Society and Industry, Special session on "Advanced remote sensing methods for a smarter and safer world" @ IEEE RTSI'17  Conference, Modena, Italy, September 11-13 2017.

11. F. Parmiggiani, M. Moctezuma-Flores, L. Guerrieri and M.L. Battagliere, "SAR analysis of the Larsen-C A-68 iceberg displacements", *Int. J. Remote Sensing* (2018),  doi: 10.1080/01431161.2018.1508921.

12. M. Moctezuma-Flores and F. Parmiggiani, "Tracking of the iceberg created by the Nansen Ice Shelf collapse", *Int. J. Remote Sensing*, (2017)  DOI:10.1080/01431161.2016.1275054.

13. M. Moctezuma Flores, F. Parmiggiani, and L. Lopez Lopez, "Measuring the Area of Antarctic Polynyas  with Cosmo-Sky-Med SAR-X Images", in *Proc. ESA Living Planet Symp. 2013,* ESA Special Publication SP-722, 2013.

14. Parmiggiani, F. & Moctezuma-Flores, M. & Morales, D.. (2009). Iceberg detection using COSMO-SkyMed satellite constellation images. Proceedings of SPIE - The International Society for Optical Engineering. 7477. 10.1117/12.836027.

15. Parmiggiani, F. & Morales, David & Moctezuma-Flores, M.. (2007). Surface signature of ocean convection in the greenland sea as detected by SAR and enhanced by statistical pattern analysis. International Geoscience and Remote Sensing Symposium (IGARSS). 879-881.

**Fig. 3.** ResponseToReviewerAppendixB

[Figure]

Figure 3. Segmentation results: (a) The binary case of K-means, image of 13 December 2017; by the proposes scheme (b): Image of 13 December 2017 and (c): Image of 18 January 2018. In (b) and (c) the detected A-68A object is displayed in the input SAR scenes. Despite the low contrast of the original scenes, in (b) and (c) the detected shape is homogeneous and without spurious pixels.

**Fig. 4.** Figure 3 (draft paper)

---

## Referee Comment (RC2) · Anonymous Referee #2 · 22 Sep 2020

The paper presents a method to automatically derive iceberg area changes and iceberg drift parameters from Sentinel-1 SAR scenes. The method consists of a contrast enhancing procedure and an automated segmentation.

This is a very technical paper, and the drift of the iceberg is merely used as a case study. In the introduction and the abstract the authors try to bring the results into the scientific perspective, but not in a satisfactory manner. I guess it is Ok to publish a technical paper in the Cryosphere, but then I would expect a discussion of how this new method does really improve the status quo (as this is not the first work on automated iceberg detection) and extend it to a bigger dataset. Is this method really better than previous ones? This is not clear from the manuscript. One example: in the caption

of figure 3 it is stated: "Even with the low contrast of the original scenes, the detection appears homogeneous with no spurious pixels". Then why bother with contrast enhancing techniques at all? Figure 2 shows an overview of different enhancement algorithms, and yes, the fuzzy one looks best in the image, but what does it improve for the segmentation process? There is no comparison shown. If the actual parameters, which have been measured, like iceberg area, path of drift and rotation of the iceberg would have been analyzed in relation to the scientific questions briefly mentioned in the introduction, like ocean currents, wind and sea ice drift, ocean temperature and so on, this would be a more suitable paper for The Cryosphere in my view. But this is not the case, there is only a display of the data, without discussing what this information could be actually used for.

The structure of the paper is not very clear, which also comes from the dilemma of trying to make a technical paper look more like a data focused one. The implementation of the segmentation process is first described in the discussion, but would rather belong to a "Methods" chapter. In the discussion I would expect evidence for the improvement this method provides in comparison to others, or the interpretation of the iceberg drift data. What is also not clear to me: Why not include more scenes, the iceberg has since been drifting further north between March 2019 and the submission date. During this time it also did lose more mass. As the method is automated this should be not such a big task. I would also recommend to include a table (maybe as an appendix) where all used scenes are listed, instead of mentioning them all in the text with just the dates. When analyzing the area loss along the iceberg drift it should be considered that the Polar Stereographic projection (I guess this is what the authors mean by stereo-polar projection) is not area preserving, or are you adapting the latitude of true scale for each scene? This is not clear from the text.

In summary I would recommend to either put more effort into interpreting the results of the drift tracking, or stay technical, but then show a more convincing summary of the benefits of the introduced method, in order to make it a suitable contribution for The

Cryosphere. In both cases it would need major rewriting, thus I do not see the benefit in going into more detail here.